# Dysfunction of the key ferroptosis-surveilling systems hypersensitizes mice to tubular necrosis during acute kidney injury

Wulf Tonnus [1,2,21], Claudia Meyer [1,2,21], Christian Steinebach [3], Alexia Belavgeni[1,2], Anne von Mässenhausen[1,2], Nadia Zamora Gonzalez[1,2], Francesca Maremonti[1,2], Florian Gembardt [1], Nina Himmerkus[4], Markus Latk[1,2], Sophie Locke[1,2], Julian Marschner[5], Wenjun Li[6], Spencer Short [7], Sebastian Doll[8], Irina Ingold[8], Bettina Proneth [8], Christoph Daniel [9], Nazanin Kabgani[10], Rafael Kramann[10,11], Stephen Motika[12], Paul J. Hergenrother [12], Stefan R. Bornstein [13,14,15,16,17], Christian Hugo[1], Jan Ulrich Becker[18], Kerstin Amann[9], Hans-Joachim Anders[5], Daniel Kreisel[6,19], Derek Pratt [7], Michael Gütschow[3], Marcus Conrad [8,20] & Andreas Linkermann [1,2 ✉]

Acute kidney injury (AKI) is morphologically characterized by a synchronized plasma membrane rupture of cells in a specific section of a nephron, referred to as acute tubular necrosis (ATN). Whereas the involvement of necroptosis is well characterized, genetic evidence supporting the contribution of ferroptosis is lacking. Here, we demonstrate that the loss of ferroptosis suppressor protein 1 (Fsp1) or the targeted manipulation of the active center of the selenoprotein glutathione peroxidase 4 (Gpx4$^{cys/-}$) sensitize kidneys to tubular ferroptosis, resulting in a unique morphological pattern of tubular necrosis. Given the unmet medical need to clinically inhibit AKI, we generated a combined small molecule inhibitor (Nec-1f) that simultaneously targets receptor interacting protein kinase 1 (RIPK1) and ferroptosis in cell lines, in freshly isolated primary kidney tubules and in mouse models of cardiac transplantation and of AKI and improved survival in models of ischemia-reperfusion injury. Based on genetic and pharmacological evidence, we conclude that GPX4 dysfunction hypersensitizes mice to ATN during AKI. Additionally, we introduce Nec-1f, a solid inhibitor of RIPK1 and weak inhibitor of ferroptosis.

---

A full list of author affiliations appears at the end of the paper.

A compelling series of publications recently indicated the involvement of necroptosis in acute kidney injury (AKI), acute tubular necrosis (ATN), the generation of necrotic casts, and the reduction of renal blood flow[1–5]. Necroptosis is detectable in human AKI[6,7]. A necroptosis-independent pathway of regulated necrosis is triggered by iron and is referred to as ferroptosis[8–11], a regulated cell death program that involves death propagation between cells in a non-random, synchronized manner[12–15]. The ferroptosis key players glutathione peroxidase 4 (GPX4)[16,17] and ferroptosis suppressor protein 1 (FSP1)[18,19] were recently demonstrated to inhibit ferroptosis in a glutathione-dependent or independent manner, respectively. The role of these proteins in AKI has not been assessed, and genetic evidence for the involvement of ferroptosis in AKI is lacking. From a therapeutical perspective, inhibitors of necroptosis and ferroptosis have been reported[20], but no single compound has been described that effectively inhibits necroptosis and ferroptosis to treat AKI. Here, we provide genetic evidence for the involvement of ferroptosis in AKI. We also introduce Nec-1f, a combined solid inhibitor of RIPK1 and a weak inhibitor of ferroptosis for the treatment of AKI and potentially other diseases.

## Results

### Genetic evidence for the contribution of ferroptosis to acute kidney injury (AKI) is lacking.

Therefore, we took advantage of a most powerful, ferroptosis-relevant mouse model, whereby the active site selenocysteine of glutathione peroxidase 4 ($Gpx4^{cys/flox}$-mice) was site-directedly mutated to its functional redox-competent analog cysteine[17]. This amino acid exchange enables GPX4 to rescue from ferroptosis[17,21], but at the same time renders the mutant enzyme highly susceptible to peroxide-induced, irreversible inactivation and consequently in a reduced resistance to ferroptosis. Tamoxifen-induced $ROSA-CreERT2-Gpx4^{cys/flox}$-mice (in the following referred to as $Gpx4^{cys/−}$-mice) do not exhibit a spontaneous renal phenotype as assessed by period acid Schiff (PAS)-staining (Fig. S1A). In a well-established model of severe bilateral kidney ischemia-reperfusion injury (IRI) that causes 50% lethality after 5 days in control mice, all $Gpx4^{cys/−}$-mice died within the first 48 h of reperfusion (Fig. 1A). In a second model of sublethal standard kidney IRI (see methods sections for details), $Gpx4^{cys/−}$-mice exhibited significantly more positivity of the lipid peroxidation marker 4HNE (Fig. S2A, B), significantly more structural damage in the kidneys (Fig. 1B, C), higher levels of serum creatinine (Fig. 1D) and higher concentrations of serum urea (Fig. 1E) 48 h following the onset of reperfusion. Higher-resolution PAS-stained kidney sections revealed massive tubular necrosis (Fig. S3) in the $Gpx4^{cys/−}$-mice. Such vesicle-like structures inside the intact tubular basal laminar (Fig. S3), which occur in kidney IRI of $Gpx4^{cys/−}$-mice have never been reported previously to the best of our knowledge. In a model of cisplatin-induced AKI, $Gpx4^{cys/−}$-mice did not exhibit any differences in standard readout systems (Fig. S4), indicating that GPX4 dysfunction does not sensitize to all forms of AKI. In conclusion, these data suggest that a replacement of the active site selenocysteine to cysteine in GPX4 results in GPX4 dysfunction and hypersensitizes mice to ferroptosis in renal tubules in kidney IRI.

The ferroptosis suppressor protein 1 (FSP1) has been recently identified as the second mainstay in ferroptosis by inhibiting ferroptosis in a glutathione-independent manner[18]. $Fsp1$-deficient mice do not exhibit a spontaneous renal phenotype at the age of 12 weeks (Fig. S1B). When subjected to kidney IRI, $Fsp1^{−/−}$-mice revealed a trend to higher levels of 4HNE (Fig. S2C, D) and significantly increased amounts of tubular necrosis were detected (Fig. 1F, G) alongside with higher serum concentrations of creatinine (Fig. 1H) and urea (Fig. 1I). These data suggest the existence of a GSH-independent anti-ferroptotic system that protects renal structure and function during IRI. In comparison to the GSH-dependent system, however, the genetic absence of $Fsp1$ resulted in less dramatic sensitization to IRI. In conclusion, the presented data add genetic evidence to the previously reported pharmacological evidence on the role of ferroptosis in AKI.

We and others recently reported on the fundamental contribution of necroptosis in AKI[2,6,22–24]. To the best of our knowledge, the pathways of necroptosis and ferroptosis are molecularly separated, but in combination contribute to the pathogenesis of IRI. Inhibitors of both pathways have been described. However, to translate several inhibitors of ferroptosis and necroptosis to clinical trials is difficult. We, therefore, set out to design a single small molecule that inhibits both RIPK1 and ferroptosis. Fig. S5A demonstrates the route of synthesis of the dual-active compound Nec-1f which was purified (Fig. S5B) and generated in sufficient amounts. Nec-1f contains a thiohydantoin moiety (Fig. 2A), which we previously demonstrated to inhibit kidney IRI[25], and an ester-bound chlorine, which binds the RIPK1 kinase pocket (Fig. S5C, D), as described in detail for the necroptosis inhibitor necrostatin-1s (Nec-1s)[26].

Given that the vast majority of small molecule inhibitors of ferroptotic cell death, particularly those containing aromatic amine moieties[27–29], are radical-trapping antioxidants (RTAs)[10], we investigated whether Nec-1f possesses this reactivity. Unlike PMC, the truncated analog of α-tocopherol, the archetype lipophilic RTA, Nec-1f displayed no activity in a cumene/STY-BODIPY co-autoxidation assay[30], even when present at 5-fold higher concentration than PMC (Fig. 2B). Nec 1f was similarly devoid of RTA activity in co-autoxidations of egg phosphatidylcholine lipids in STY-BODIPY-embedded liposomes[29], in stark contrast to both PMC and ferrostatin-1 (Fer-1), the prototype ferroptosis inhibitor (Fig. 2C, D), confirming that Nec-1f does not function as an RTA. To further characterize Nec-1f, we performed an off-target analysis using 10 μM of Nec-1f (Fig. S6A). In that assay, inhibition of 50% was reached for the glucocorticoid receptor (GR). We, therefore, knocked down the GR in HT1080 cells. Upon ferroptosis-induction using erastin for 24 h, GR knockdown did not result in a significant difference with respect to a classical FACS readout (Fig. S6B), suggesting no relevant off-target effect of Nec-1f with respect to the investigated enzymes.

We next tested 10 μM Nec-1f alongside Nec-1s and Fer-1 in HT29 cells, which were induced to undergo necroptosis with the combination of TNFα, the smac mimetic birinapant, and the caspase inhibitor zVAD-fmk (referred to as TSZ treatment). As demonstrated in Fig. 2E, Nec-1s and Nec-1f maintained double negativity in FACS assays for 7-AAD and annexin V, whereas Fer-1 did not provide protection against necroptosis. Mechanistically, Nec-1f prevented phosphorylation of mixed-lineage kinase domain-like (MLKL, pS358), the downstream mediator of necroptosis in human cells (Fig. 2F). Nec-1s and Nec-1f, due to the nature of inhibiting RIPK1 kinase activity, also prevented phosphorylation at S166 in RIPK1 (Fig. 2G). In this experiment, we added Fer-1 as a control compound and found that RIPK1 S166 phorphorylation was increased in the presence of zVAD-fmk, indicating a caspase-independent unknown way of boosting RIPK1 phosphorylation. Interestingly, the efficacy of inhibition of MLKL-phosphorylation was comparable between Nec-1s and Nec-1f, while Fer-1 did not prevent MLKL-phosphorylation (Fig. 2H). These effects were not limited to human cells, as murine NIH3T3 treated with TNFα and zVAD-fmk phenocopied the response of HT29 cells (Fig. 2I). These data demonstrate that Nec-1f indeed inhibits RIPK1 kinase activity, resulting in the prevention of downstream phosphorylation of MLKL.

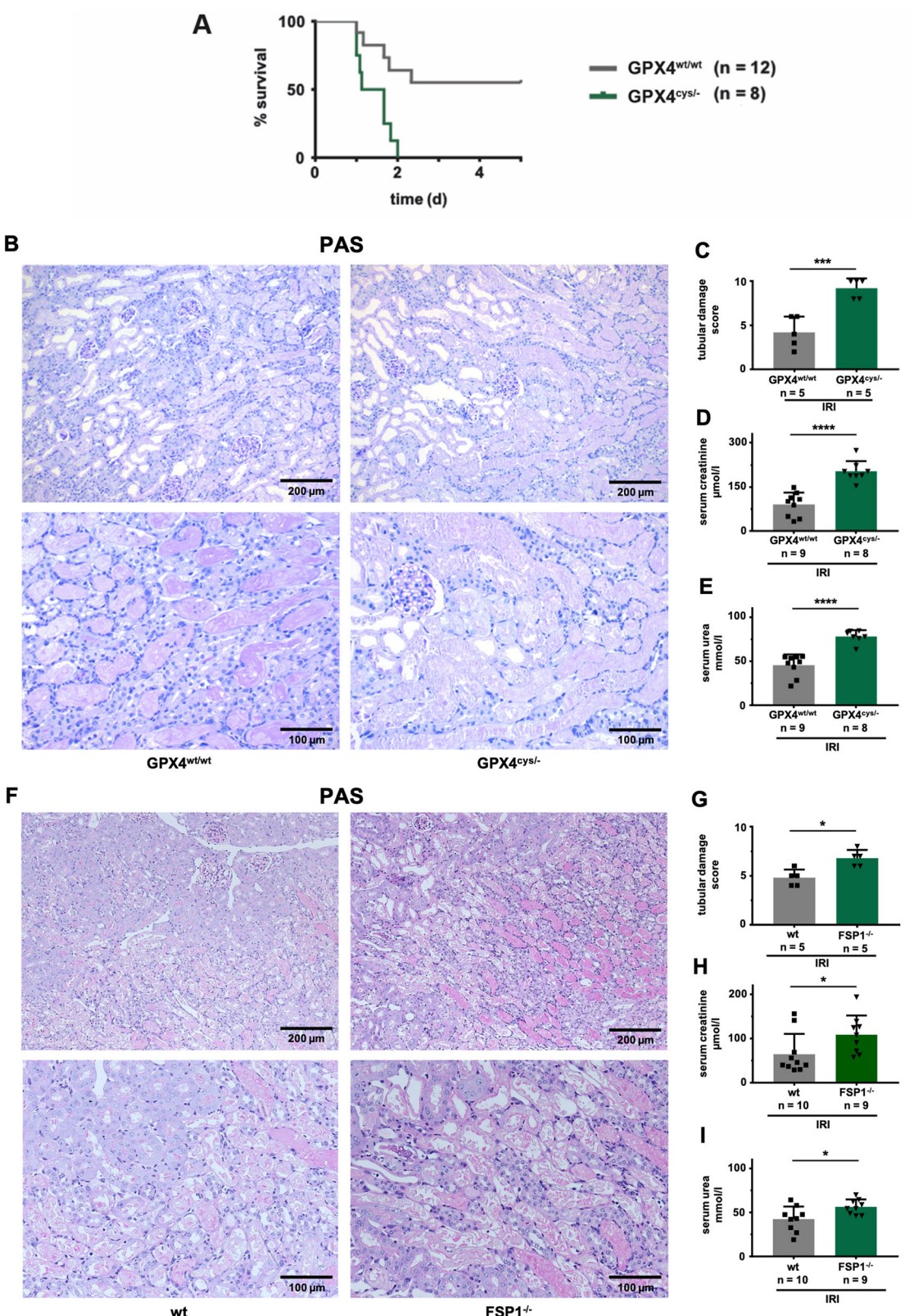

Following the observation that Nec-1f, unlike Fer-1, is not an RTA, we investigated the antiferroptotic effect of Nec-1f in comparison to Fer-1 and Nec-1s. Four classes of ferroptosis inducers (FINs) have been described (erastin (type 1), RSL3 (type 2), FIN56 (type 3), and FINO2 (type 4))[31]. In addition, recent data suggested that the thioredoxin-reductase inhibitor ferroptocide might function in a related way[32]. We, therefore, established dose-response curves for the four FINs and ferroptocide (Fig. S7A, B) in human HT1080 cells and identified the inhibitory properties of Fer-1, Nec-1s, and Nec-1f in these settings (Fig. S7C). Fig. 3A shows a representative experiment (quantified in Fig. S8A) and demonstrates inhibition of cell death induction by Fer-1 and Nec-1f, but not by Nec-1s. We noticed that Nec-1f is less effective in preventing FINO2-induced

**Fig. 1 Genetic evidence for the contribution of ferroptosis to acute kidney injury. A** Loss of the selenocysteine residue of GPX4 sensitizes mice to acute tubular ferroptosis. Fifthteen days after induction of the Cre-driver, severe renal ischemia-reperfusion injury (IRI) was performed in $Gpx4^{cys/-}$ mice, and overall survival is presented in a Kaplan–Meier plot. **B** Mice were induced as in (**A**), and standard IRI was performed. Forty-eight hours after the onset of reperfusion, **B** representative microphotographs of periodic acid-Schiff (PAS)-stained histological sections of IRI-treated mice are presented and the **C** tubular damage score was quantified. Serum levels of creatinine (**D**) and urea (**E**) were detected. **F** IRI-treated wild-type C57Bl/6 mice and *Fsp1*-deficient mice underwent standard renal IRI. Forty-eight hours following the onset of reperfusion, representative PAS-stained histological sections are presented and the tubular damage score was quantified (**G**). Serum concentrations of creatinine (**H**) and urea (**I**) were measured. Bar graphs represent the mean $+/-$ SD. * $p < 0.05$, *** $p < 0.001$, ****$p < 0.0001$.

ferroptosis, and that FIN56 does not effectively drive ferroptosis in this system. We set up a similar experiment with NIH3T3 cells and obtained comparable results (Fig. 3B and Fig. S8B).

NIH3T3 cells are the only cells in our hands that are sensitive to the induction of necroptosis (by TZ) and ferroptosis (e.g., by RSL3). We, therefore, induced both ferroptosis and necroptosis simultaneously in NIH3T3 cells. While NIH3T3 cells died with any of the given treatments, Nec-1f maintained cell viability in this model (Fig. 3C). These data indicate that Nec-1f simultaneously inhibits RIPK1 and ferroptosis in NIH3T3 cells.

Having confirmed the function of Nec-1f in standard cell culture assays often used for research on regulated necrosis, we continued to assess kidney cells. It was previously suggested that cell death propagation in ferroptosis involves a "wave-of-death"-like phenotype[33,34]. The non-random nature of RSL3-induced necrosis of primary renal tubular cells is demonstrated in Supplementary Movie S1 and Fig. 4A. We found the murine tubular cell line MCT to be sensitive to RSL3 and FINO2-induced ferroptosis (Fig. S9A), and this effect was reversed by Fer-1. While Nec-1f failed to prevent the FINO2-induced death, it inhibited RSL3-induced annexin V / 7-AAD positivity in MCT cells (Fig. S9A). A recently generated human kidney tubular cell line (CD10-135, provided by Rafael Kramann) exhibited sensitivity to all FINs, and LDH release from CD10-135 cells was reversed by Fer-1 or Nec-1f (Fig. 4B and Fig. S9B). Interestingly, the RTA Fer-1, but not Nec-1f, reversed the phenotype induced by ferroptocide (Fig. 4B and Fig. S9B). The non-random nature of cell death propagation, in our hands, was detected in primary renal tubular cells and the associated LDH-release was prevented by the addition of Fer-1 (Fig. 4C) or Nec-1f (Fig. 4D). We noted a typical morphological appearance upon RSL3 treatment in primary kidney tubular cells that was reversed by inhibition of ferroptosis (Fig. S10A). In conclusion, these data suggest that renal tubular cell lines are sensitive to ferroptosis, and that ferroptosis of primary renal tubular cells is mediated by ferroptosis in a "wave-of-death"-like manner.

Using a standardized protocol to freshly isolate primary renal tubules (Fig. S11), we confirmed the unique pattern of cell death propagation in renal tubules (Supplementary Movie S2 and Fig. 4E). Importantly, in the case of freshly isolated kidney tubules, this cell death occurs spontaneously (Fig. 4E and Supplementary Movie S2). In previous data[14], the addition of 5 mM glycine diminishes the LDH release, but an osmotic effect on the readout assay cannot be ruled out in that case. We, therefore, investigated LDH release of kidney tubules in the absence of glycine (Fig. 4F). Nevertheless, even in conditions where glycine was added to the media, freshly isolated primary tubules clearly exhibited higher amounts of LDH release after 2 h of treatment with RSL3 (Fig. 4G and Fig. S10B), and this effect was significantly attenuated by Fer-1 or Nec-1f (Fig. 4G). Importantly, when mice were injected with 2 mg/kg of either Fer-1 or Nec-1f 30 min before the kidneys were removed and tubules were isolated in glycine-containing media, less spontaneous LDH release was measured (Fig. 4H). Collectively, these data indicate that ferroptosis drives tubular necrosis and LDH release from kidney tubules and that Fer-1 or Nec-1f potently inhibits this phenomenon.

Given the results of Nec-1f as a tool to inhibit RIPK1 and ferroptosis in cell lines, primary kidney cells, and kidney tubules, we decided to investigate hepatic microsome stability and human plasma stability (Fig. 5A, B and Fig. S12A) and the tissue PK of Nec-1f in plasma, whole blood, brain, kidney, heart (Fig. S12B), liver and lung (Fig. S12C). Assessment of PK parameters revealed a kidney tissue half-life of 211 min for Nec-1f (Fig. S12D). We decided to test Nec-1f in murine models of acute kidney injury and after cardiac transplantation.

Necroptosis-deficient mice were recently demonstrated to be protected in a mouse model of Ca-oxalate (CaOx)-induced toxic AKI[2,35]. We treated mice with either vehicle or 1.65 mg/kg body weight Nec-1f. While CaOx crystal deposition was comparable in both groups (Fig. 5C), Nec-1f reduced tubular injury (Fig. 5D). Along similar lines, the number of TUNEL-positive cells was significantly decreased by Nec-1f (Fig. S13). Functionally, mice exhibited lower concentrations of serum urea (Fig. 5E) and serum creatinine (Fig. 5F) upon Nec-1f treatment. These data suggest that Nec-1f protects mice from a necroptosis-driven form of AKI.

We next set out to investigate whether Nec-1f can ameliorate inflammatory responses related to IRI after transplantation of a solid organ. We recently reported that ferroptosis triggers neutrophilic graft infiltration after heart transplantation[36]. Treatment of recipient mice with Nec-1f (2.0 mg/kg body weight) at the time of transplantation inhibited the recruitment of neutrophils after reperfusion of the graft (Fig. 5G–K and Supplementary Movies S3, S4). While Nec-1f did not alter the rate at which neutrophils were recruited to the graft, we observed significant increases in their rolling velocities, highly significant decreases in their density, and highly significant reductions in their extravasation when compared to grafts that were transplanted into control recipients. Thus, similar to our previous observations with Fer-1, Nec-1f substantially blunts neutrophil recruitment to transplanted hearts[36].

We next pretreated mice with Nec-1f (1.65 mg/kg body weight) in a single dose intraperitoneally 15 min prior to induction of bilateral kidney IRI. As demonstrated in Fig. 6A, 4HNE staining of the renal tissue was attenuated (quantified in Fig. 6B). The typical appearance of tubular necrosis post IRI was less pronounced (Fig. 6C). In keeping with this finding, levels of tubular damage (Fig. 6D) and serum levels of creatinine and urea were lower than in the vehicle-treated mice (Fig. 6E, F). In line with the recent concept of necroinflammation in AKI[37,38], immunohistochemical staining for a macrophage marker (F4/80) detected significantly less infiltration of macrophages in the renal medulla (Fig. 6G–I). Finally, we performed an independent experiment employing the model of severe renal IRI (see "Methods" for details) to directly compare the effects of 2.0 mg/kg body weight of Fer-1, Nec-1s, and Nec-1f, respectively. As presented in the Kaplan–Meier plot in Fig. 6J, the percentage of surviving mice was significantly higher after a single dose of Nec-1f prior to the onset of reperfusion.

## Discussion

The inactivation of GpX4 in mice drives chronic renal injury[39], but the involvement of this system in a clinically important

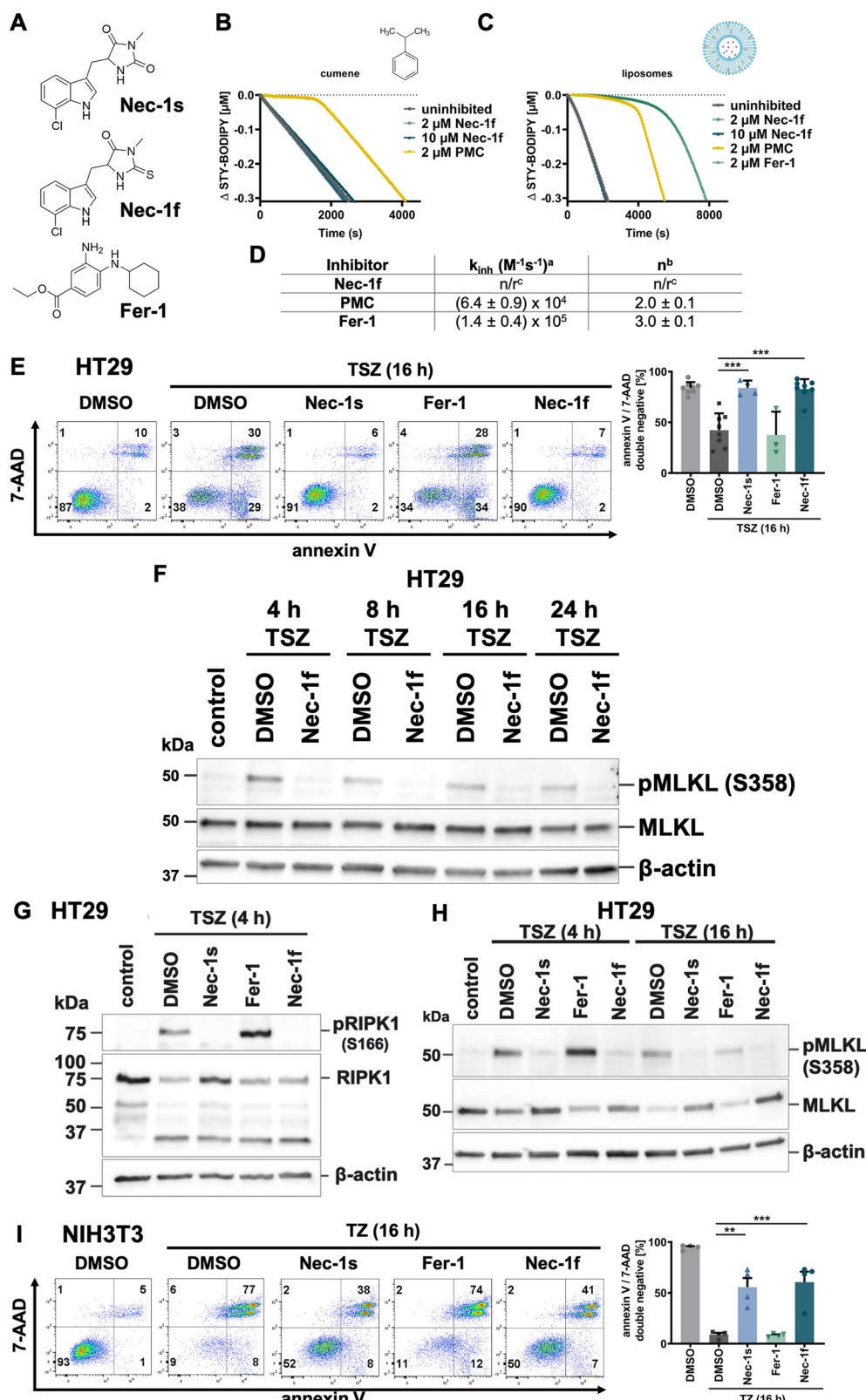

model of AKI was not investigated. Here, we provide genetic evidence for the importance of this system in AKI and solid organ transplantation. It is entirely unclear to what extent the ubiquinone system involving FSP1 might compensate for GpX4 dysfunction. Future studies will address this question by generating combined Gpx4$^{cys/-}$-Fsp1-deficient mice. However, the overwhelming data on the contribution of necroptosis to AKI in combination with the growing body of evidence for ferroptosis raise important questions: Are these two cell death systems independent in the complex setting of diseases such as AKI and solid organ transplantation? What is the relative contribution of each pathway? To the best of our knowledge, no convincing piece of evidence exists so far to demonstrate an interregulation.

**Fig. 2 Nec-1f inhibits RIPK1-dependent necroptosis. A** Structures of small molecules used in this study. **B** Representative co-autoxidations of cumene and STY-BODIPY initiated by Azobisisobutyronitrile (AIBN) and carried out in the presence of Nec-1f or 2,2,7,8-pentamethyl-6-chromanol (PMC). **C** Representative co-autoxidations of STY-BODIPY and the polyunsaturated fatty acids of egg phosphatidylcholine liposomes initiated by MeOAMVN and carried out in the presence of Nec-1f, PMC, and Fer-1. **D** Inhibition rate constants ($k_{inh}$) and stoichiometries (n) derived from the initial rates and lengths of the inhibited periods, respectively, of the data in panel (**C**). **E** HT29 cells were treated for 16 h with TNFα (20 ng/ml), birinapant (1 μM), and zVAD-fmk (20 μM) (TSZ) for the induction of necroptosis. The influence of small molecule inhibitors on necroptosis induction is shown by 7-aminoactinomycin (7-AAD) and annexin V positivity. (n = 8 for DMSO, TSZ and Nec-1f; n = 4 for Nec-1s and Fer-1) **F** HT29 cells were treated for indicated times with TSZ and simultaneous Nec-1f (10 μM) treatment. pMLKL (S358) positivity is shown by western blotting, MLKL and β-actin serve as loading controls. **G, H** HT29 cells were treated for indicated times with TSZ in the presence of Nec-1s (10 μM), Fer-1 (1 μM) and Nec-1f (10 μM). pMLKL (S358) and pRIPK1 (S166) are demonstrated. **I** NIH3T3 cells were treated with TZ for 16 h and stained for 7-AAD and annexin V (n = 4). The bar graph shows mean +/− SD. Statistical analysis was performed using one-way ANOVA (post hoc Tukey's). ** $p < 0.01$, *** $p < 0.001$.

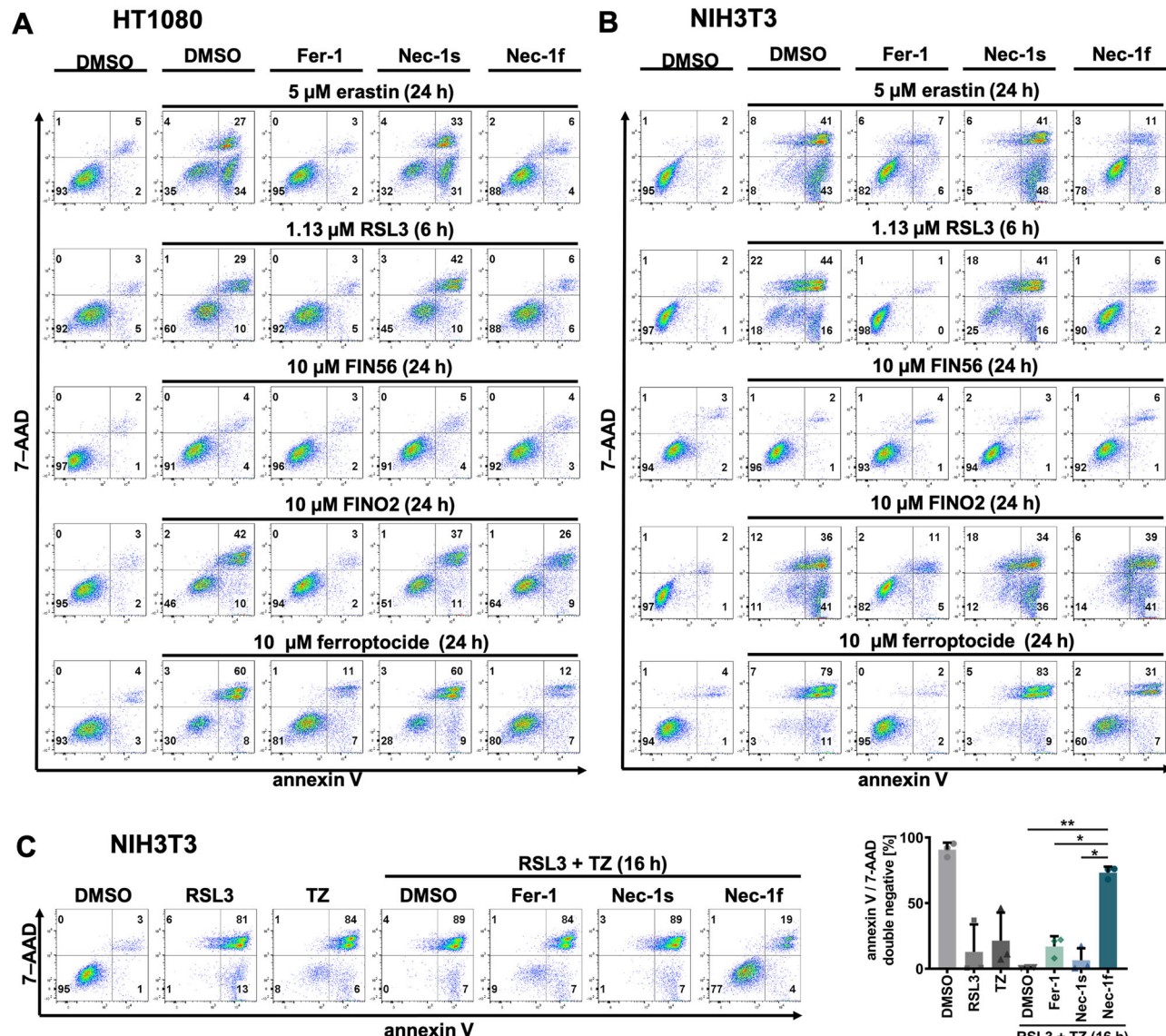

**Fig. 3 Nec-1f inhibits ferroptosis. A** HT1080 cells were treated for 6 or 24 h with ferroptosis inducers (FINs) or ferroptocide in the presence of Fer-1 (1 μM), Nec-1s (30 μM) or Nec-1f (30 μM), respectively, as indicated. **B** NIH3T3 cells were treated as in (**A**). **C** NIH3T3 cells were treated for 16 h with either 1.13 μM RSL3 alone, TZ alone, or the combination of RSL3 plus TZ. While Fer-1 or Nec-1s fail to rescue NIH3T3 cells from necrosis, Nec-1f does. Plots show stainings of 7-aminoactinomycin (7-AAD) and annexin V (n = 3). The bar graph shows mean +/− SD. Statistical analysis was performed using one-way ANOVA (post hoc Tukey's). * $p < 0.05$, ** $p < 0.01$.

The generation of combined inhibitors to target necrotic cell death comes with several advantages. First, compensation mechanisms, such as those potentially common in multiple protein-deficient murine systems, are negligible upon short-term application. Second, drug interactions can be limited. Finally, and most importantly, in light of potential translation into first-in-class clinical trials, a single small molecule will be more readily approved. Lower costs and efforts in conducting regulatory safety

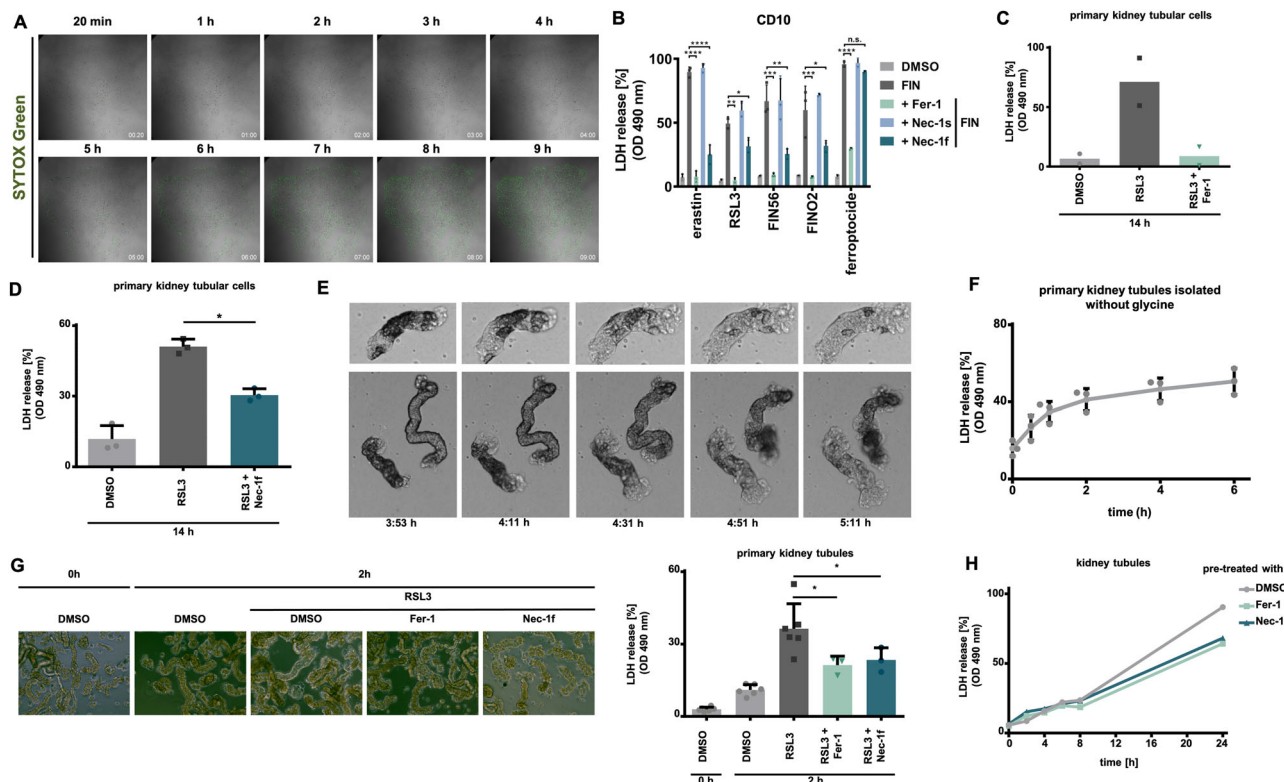

**Fig. 4 Ferroptosis in primary tubular cells and in freshly isolated murine renal tubules is prevented by ferroptosis inhibitors. A** Still images of a time-lapse video (Supplementary movie 1) of primary murine renal tubular cells demonstrating cell death propagation (SYTOX™ Green) following ferroptosis induction with 1.13 μM RSL3. **B** The human renal tubular cell line CD10-135 was treated for indicated times with type 1–4 ferroptosis inducers (erastin, RSL3, FIN56, FINO2) or ferroptocide in the presence of Fer-1, Nec-1s, or Nec-1f. Lactate dehydrogenase (LDH) release is presented (n = 3). **C** Primary kidney tubular cells were treated with RSL3 for 14 h. Note the complete reversal of LDH release by Fer-1 (n = 2). **D** Primary kidney tubular cells were treated with RSL3 for 14 h in the presence of Nec-1f (n = 3). **E** Still images and magnifications of Supplementary movie 2 demonstrating characteristic morphological changes ("wave of death"-like cell death progression) in freshly isolated renal tubules that were left unstimulated. **F** LDH release as a function of time of freshly isolated primary kidney tubules that were left untreated in standard medium (n = 3). **G** LDH release from unstimulated freshly isolated primary kidney tubules in the presence of 5 mM glycine (DMSO, 0 h, DMSO 2 h and RSL3 2 h n = 6, Fer-1 and Nec-1 n = 3). **H** Freshly isolated renal tubules were cultured for 2 h in a glycine-containing medium in the presence of RSL3 in the presence of Fer-1 or Nec-1f. Representative images are presented and the LDH-release was quantified. Note the reduction of LDH release in the presence of Nec-1f. **H** Freshly isolated primary kidney tubules were kept in 5 mM glycine-containing medium. LDH release over time is demonstrated. The addition of Nec-1f or Fer-1 resulted in less LDH release at the 24-h time point. All experiments shown are representative of two to five independent complete repetitions performed. The bar graphs show mean +/− SD. Statistical analysis was performed using one-way ANOVA (post hoc Tukey's). * p < 0.05, ** p < 0.01, *** p < 0.001, **** p < 0.0001. Thirty micromolar Nec-1f was used in all panels in this figure.

and efficacy studies compared to what is required for combination therapies will prioritize trials employing combined inhibitors. However, our data suggest that while Nec-1f is a potent inhibitor of RIPK1, the concentrations required to effectively inhibit ferroptosis are high (30 μM in cell culture, 2.0 mg/kg body weight in murine injury models). In addition, with respect to its function as a ferrostatin, we failed to identify the molecular target of Nec-1f and thus, the mechanism of action. Further modification of Nec-1f, including the anti-ferroptosis potency, and the identification of the molecular mechanism of action will be required before a dual inhibitor may be tested in clinical trials.

In summary, based on two lines of genetic and pharmacological evidence, we conclude that ferroptosis contributes to acute kidney injury induced by IRI. In addition, we introduce Nec-1f as a novel tool reagent to simultaneously inhibit RIPK1 and ferroptosis.

## Methods
**Cell lines and cell culture**. Murine NIH3T3 (Cat# CRL-1658) and human HT1080 (Cat# CCL-121™) cell lines were purchased from the American Type Culture

Collection. Mouse cortical tubule cells (MCT) were kindly provided by Alberto Ortiz, HT29 cells were kindly provided by Simone Fulda. Immortalized human kidney tubular epithelial CD10-135 cells were established in the Rafael Kramann laboratory and kindly provided for this project. Cells were cultured in a humidified 5% $CO_2$ atmosphere. HT1080, HT29, NIH3T3, and MCT cell lines were cultured in Dulbecco's Modified Eagle Medium (DMEM, Thermo Fisher) supplemented with 10% (v/v) FBS (Thermo Fisher), 100 U/ml penicillin, and 100 μg/ml streptomycin (Pen/Strep, Thermo Fisher). The CD10-135 cells were cultured in DMEM:F12 Glutamax (Thermo Fisher) supplemented with 10% FBS and Pen/Strep.

**Cell death assays**. To induce necroptosis in a human cell line, HT29 cells were stimulated at the indicated time points with 20 ng/ml human TNFα (Biolegend), 1 μM SMAC-mimetics (Birinapant, Chemietek), and 20 μM zVAD-fmk (Selleckchem, herein referred to as zVAD). For the induction of necroptosis in NIH3T3 cells, cells were incubated with 20 ng/ml TNF-α and 20 μM zVAD for indicated time points. Ferroptosis was induced using four ferroptosis inducers (FINs): Type 1 FIN: erastin (Sigma Aldrich), type 2 FIN: RSL3 (Selleckchem), type 3 FIN: FIN56 (Sigma Aldrich), type 4 FIN: FINO2 (provided by Keith Wörpel & Brent Stockwell). Necrosis was also induced as described[32] using the thioredoxin reductase inhibitor ferroptocide (provided by Paul J. Hergenrother). Cells were seeded in six-well plates or petri dishes. Unless otherwise indicated, we used 5 μM erastin, 1.13 μM RSL3, 10 μM FIN56, 10 μM FINO2, and 10 μM ferroptocide (also see Supplementary Table 2). After indicated time points, cells were collected and prepared for flow cytometry, immunoblotting, and/or LDH release assays.

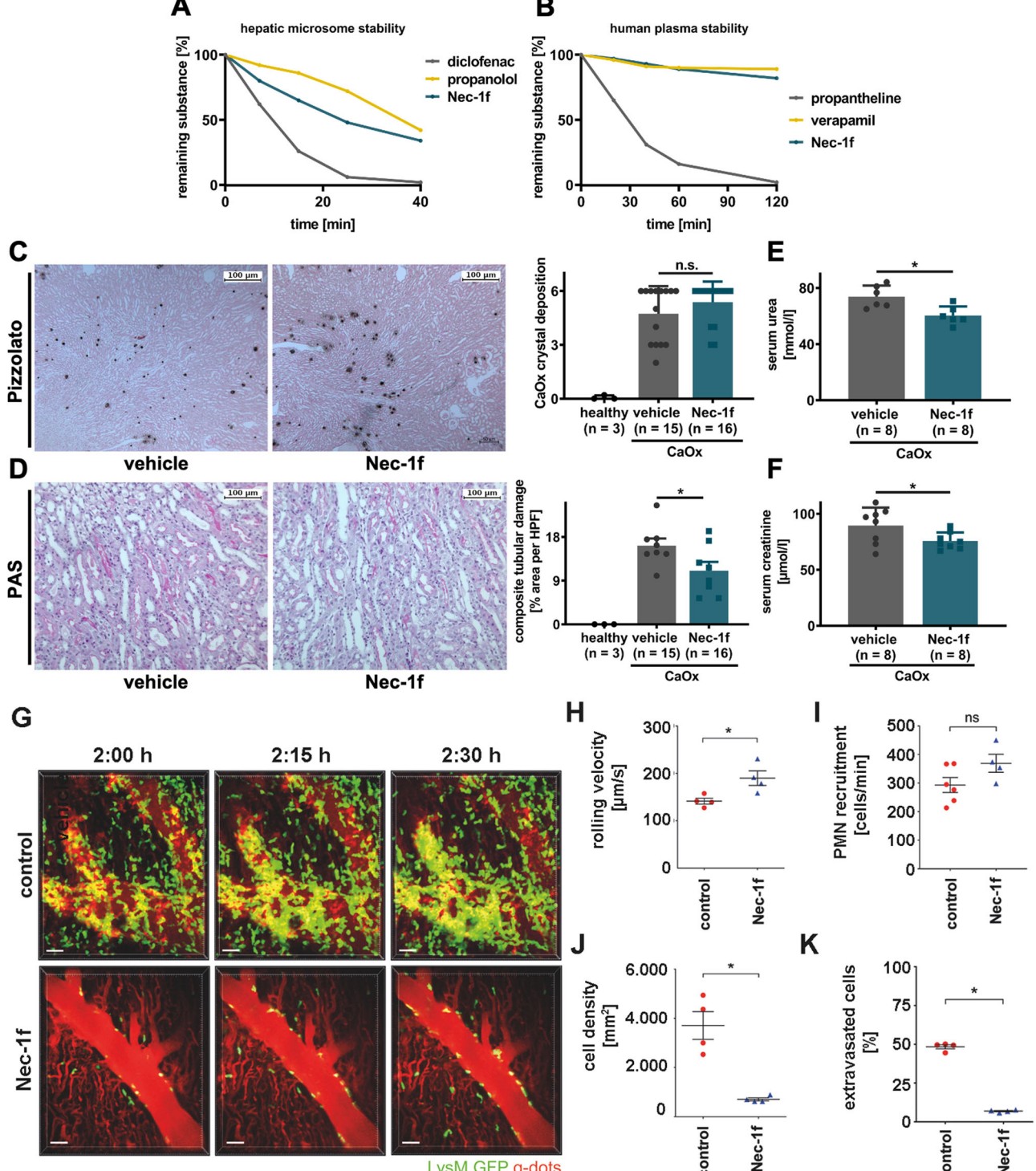

**Fig. 5 Nec-1f protects from calcium oxalate (CaOx)-induced nephropathy and reduces neutrophil recruitment after heart transplantation. A** Human microsomal stability for Nec-1f in comparison with reference substances diclofenac and propanolol. **B** Human plasma stability of Nec-1f was assessed in comparison with reference substances (propantheline, verapamil) over 120 min. **C** Mice received a single injection of 100 mg/kg sodium oxalate and were supplemented with 3% sodium oxalate in the drinking water upon treatment with the vehicle of 1.65 mg/kg Nec-1f. Pizzolato staining visualizes calcium oxalate crystal deposition in the kidneys. Quantification of CaOx crystal deposition demonstrated equal amounts of deposition in each group. **D** Periodic acid-Schiff (PAS) staining in kidneys after CaOx treatment was used to assess the tubular damage. **E** Serum urea and **F** serum creatinine levels were analyzed 24 h after the injection of CaOx. The bar graphs (**C**–**F**) show mean +/− SD. Statistical analysis was performed using two-sided students t-test. **G** Two-photon intravital imaging of neutrophil (green) behavior after transplantation of B6 hearts into control untreated (top) or Nec-1f-treated (2 mg/kg) (bottom) B6 LysM-GFP mice at indicated time points after reperfusion. Vessels were labeled red after the injection of quantum dots (q-dots). $n = 4$ per experimental group. Scale bars: 30 μm. **H** Intravascular rolling velocities of neutrophils, **I** neutrophil recruitment per minute to coronary veins, **J** density of neutrophils, and **K** percentage of extravasated neutrophils in control cardiac grafts and after treatment of recipient mice with Nec-1f. **H**–**K** Shows the mean +/− SEM. *$p < 0.05$; n.s. not significant. Statistics were performed using the Mann−Whitney U test.

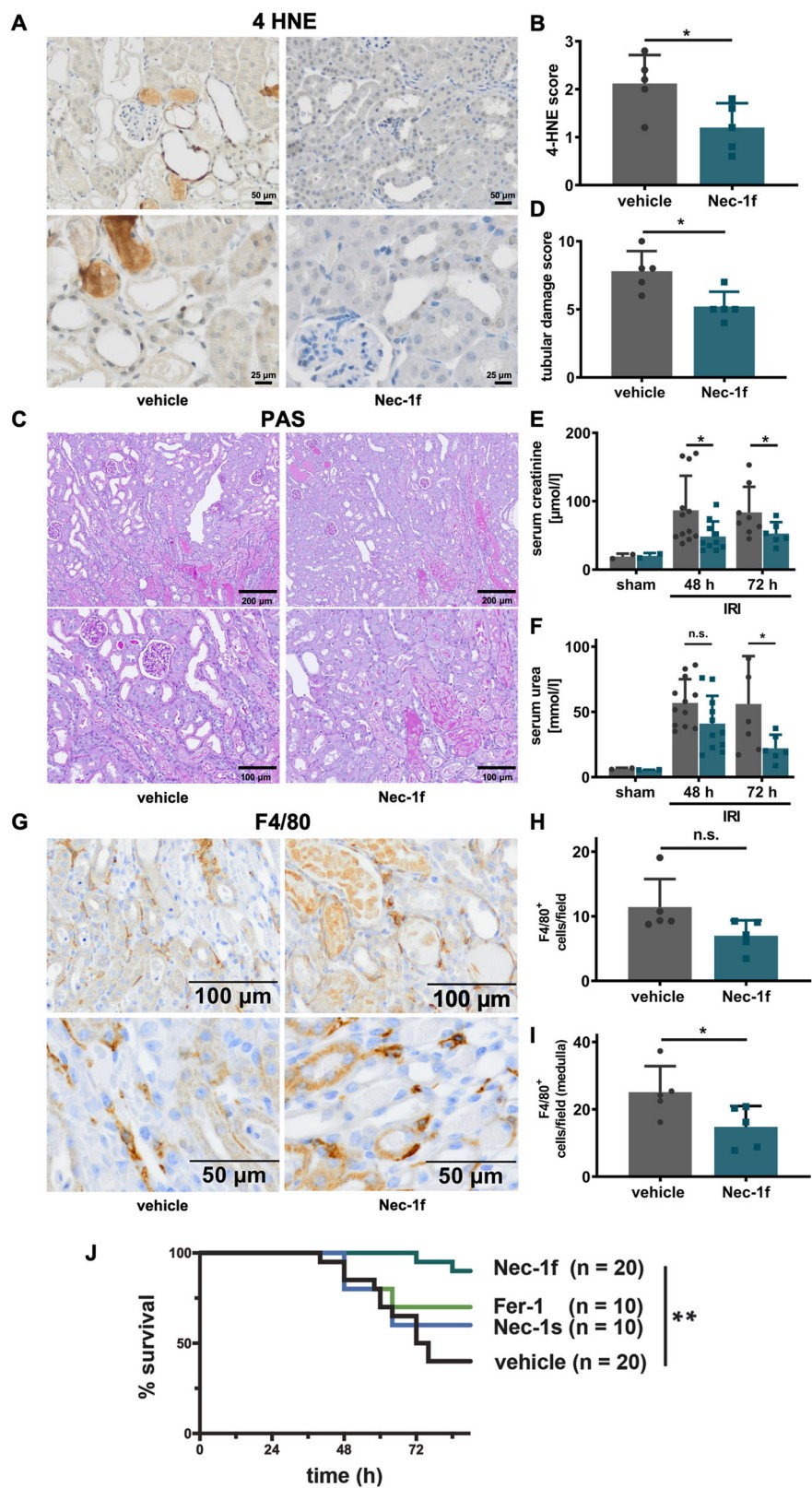

**Western blotting**. Cells were lysed in ice-cold 10 mM Tris-HCl, pH 7.5, 50 mM NaCl, 1% Triton X-100, 30 mM sodium pyrophosphate, 50 mM NaF, 100 μM Na$_3$VO$_4$, 2 μM ZnCl$_2$, and 1 mM phenylmethylsulfonyl fluoride (PMSF, modified Frackelton buffer) for 30 min on ice. Insoluble material was removed by centrifugation (14 000×$g$, 30 min, 4 °C). Protein concentration was determined using a commercial Bradford assay kit according to the manufacturer's instructions (Thermo Fisher). Equal amounts of protein (typically 30 μg per lane) were resolved on a 4%−15% gradient SDS/PAGE gel and transferred to a PVDF membrane (BIO-RAD). Primary antibody incubation was performed for anti-pS166-RIPK1 (Cell Signaling, Cat# 65746, 1:1000 dilution), anti-pS358-MLKL, (Abcam, ab187091, 1:1000 dilution), anti-MLKL (Genetex, GTX107538, 1:1000 dilution) and anti-β-Actin (Cell Signaling, Cat# 3700S, 1:1000 dilution). Secondary antibodies (anti-mouse, HRP-linked antibody, Cell Signaling, Cat# 7076S; anti-rabbit, HRP-linked antibody, Cell Signaling, Cat# 7074S) were applied at concentrations of

**Fig. 6 Nec-1f protects against bilateral renal ischemia-reperfusion injury in mice.** C57Bl/6 N mice underwent bilateral ischemia as detailed in the methods section. **A** Representative images of 4-Hydroxynonenal (4HNE)-staining, quantified in (**B**, $n = 4$). **C** Representative periodic acid-Schiff (PAS)-stained histological sections are presented and quantified using the tubular damage score (**D**, $n = 4$). Serum concentrations of **E** creatinine (sham $n = 2$, 48 h IRI vehicle $n = 12$, Nec-1f $n = 10$, 72 h IRI vehicle $n = 8$, Nec-1f $n = 6$) and **F** urea (sham $n = 2$, 48 h IRI vehicle $n = 12$, Nec-1f $n = 11$, 72 h IRI vehicle $n = 6$, Nec-1f $n = 6$) were measured in two independent experiments 48 and 72 h following the onset of reperfusion. **G** F4/80 positivity was stained by immunohistochemistry and quantified by experienced nephropathologists (**H**, **I**, $n = 5$). Bar graphs represent the mean +/− SD. **J** Kaplan−Meier survival curves following severe renal ischemia-reperfusion injury upon pre-treatment with 2 mg Fer-1/kg body weight, 2 mg Nec-1s/kg body weight, 2 mg Nec-1f/ kg body weight, or vehicle control are shown. Differences in the survival curves of vehicle compared to Fer-1 or Nec-1s are not statistically significant. All experiments were performed in a strictly double-blinded manner. Statistical analysis was performed using a two-sided student's t-test. *$p < 0.05$; n.s. not significant.

1:5.000 (also see Supplementary Table 1 for details). Proteins were then visualized by enhanced chemiluminescence (ECL; Amersham Biosciences).

**Flow cytometry**. Cells were harvested and the pellets were washed twice in PBS and stained with 5 µl of 7-AAD (BD Biosciences) and 5 µl of annexin-V-FITC (BD Biosciences) added to 100 µl annexin-V binding buffer (BD Biosciences). After 15 min, cells were recorded on the LSRII with the FACS Diva 6.1.1 software (BD Biosciences) and subsequently analyzed with the FlowJo v10 software (Tree Star).

**LDH release assay**. The LDH release of cells or of freshly isolated kidney tubules was measured according to manufacturers' instructions at indicated time points. In brief, an aliquot of the supernatant was taken and Lysis Solution was added for 45 min to induce maximal LDH release before another aliquot of the supernatant was taken. Subsequently, the supernatants were incubated with CytoTox 96® Reagent for 15 min protected from light at room temperature before adding Stop Solution. Absorbance was measured at 490 nm.

**siRNA treatment in HT1080 cells**. HT1080 cells were plated in a petri dish in 15 ml antibiotic-free medium. After 24 h 180 pmol RNAi and 24 µl Lipofectamine™ (Thermo Fisher) were each mixed in 1.5 ml Opti-MEM® I Medium (Thermo Fisher) without serum, combined and incubated at room temperature for 20 min before dropping the mixture on the cells. The following day, cells were replated into six-well-plates and treated with ferroptosis inducers and/or inhibitors after 24 h and subsequently analyzed via FACS.

**CellTiter-Glo® luminescent cell viability assay**. Cells were plated in an opaque-walled 96-well plate and treatment with ferroptosis inducers and/or ferroptosis inhibitors was initiated 24 h later. After the established treatment schedule, the CellTiter-Glo® Luminescent Cell Viability Assay (also see Supplementary Table 2 for details) was used to quantify the amount of ATP in each well to detect the presence of metabolically active cells. Here, the reaction agent was directly added to the wells, mixed on an orbital shaker for 2 min, and then incubated at room temperature for 10 min. The luminescence was detected using a Promega GloMax® instrument.

**Time-lapse imaging**. Time-lapse imaging was performed using a 2.5×/0.12 Fluar objective for the primary murine tubular cells and a 5×/0.25 Fluar objective for the primary murine renal tubules on an Axiovert 200M equipped with a large incubation chamber (37 °C), 5% CO₂, and humidity control. Transmitted light and fluorescent images (LED 475 nm os Spectra X light source, emission filter BP 525/ 50) were acquired using an Orca flash 4.0 camera. Primary murine renal tubules were isolated without glycine in any of the solutions and transferred in a six-well plate containing their respective medium as mentioned in the section "Isolation of primary murine renal tubules". Primary murine tubular cells that were passaged once were treated with RSL3 diluted in their respective medium as mentioned in the section "Generation, cell culture and induction of cell death in primary murine tubular cells". Additionally, 50 nM SYTOX Green nucleic acid stain (Life Technologies) was added to visualize the necrotic primary murine tubular cells. The live imaging procedure was supported by the Light Microscopy Facility, a Core Facility of the CMCB Technology Platform at TU Dresden.

**Assessment of radical trapping antioxidant properties**

*Inhibited co-autoxidations of cumene*. Cumene was washed three times with 1 M NaOH, distilled H₂O, and brine as previously established[30]. The organic fraction was dried using MgSO₄, filtered, and distilled under reduced pressure. The distillate was then purified by percolation through silica, followed by basic alumina. To a cuvette, 1.25 ml cumene (3.6 M) and 1.18 ml PhCl were added and equilibrated at 37 °C for 15 min in the thermostatted sample holder of a UV−Vis spectrophotometer. STY-BODIPY (12.5 µl of a 2 mM stock solution in 1,2,4-trichlorobenzene) and AIBN (50 µL of 0.3 M stock solution in PhCl) were added and the solution was thoroughly mixed. After 45 min, the test compound (10 µL of 500 µM or 2.5 mM stock solution in PhCl) was added and the loss of absorbance was

monitored at 571 nm ($\varepsilon_{\text{STY-BODIPY}} = 128, 141\ \text{M}^{-1}\ \text{cm}^{-1}$). Experiments were performed in triplicate using HPLC grade solvents and kinetic data is given as an average of three independent measurements.

*Inhibited co-autoxidation of egg phosphatidylcholine liposomes*. Based on a previously published method[27–29], to a black 96-well polypropylene plate, a solution of egg phosphatidylcholine liposomes (1 mM), STY-BODIPY (1 µM), and MeOAMVN (0.2 mM) in PBS at pH 7.4 were added to a final volume of 295 µl. Inhibitors were then added (5 µl aliquots) at desired concentrations to the appropriate wells followed by vigorous mixing. The plate was equilibrated to 37 °C and data was collected using a BioTek Synergy H1 plate reader by excitation of the probes at 488 nm and emission was measured at 518 nm. The raw data were transformed by dividing the obtained RFU values by the response factor of $2.90 \times 10^4$ RFU/µM (specific to gain 70 on the aforementioned instrument). Experiments were performed in triplicate using HPLC grade solvents and kinetic data is given as an average of three independent measurements. Experiments were performed in triplicate using HPLC grade solvents and kinetic data is given as an average of three independent measurements.

**Assessment of metabolic stability in human liver microsomes**. To determine metabolic stability in human liver microsomes (pooled, mixed gender; XenoTech, H0630/lot N# 1610016) Nec-1f was assessed using HPLC-MS. Metabolic stability was defined as the percentage of parent compound lost over time in the presence of a metabolically active test system. For this, microsomal incubations were carried out in 96-well plates in five aliquots of 40 µl each (one for each time point). Liver microsomal incubation medium comprised of phosphate buffer (100 mM, pH 7.4), MgCl₂ (3.3 mM), NADPH (3 mM), glucose-6-phosphate (5.3 mM; Sigma-Aldrich), glucose-6-phosphate dehydrogenase (0.67 units/ml, from baker's yeast, type XV; Sigma-Aldrich) with 0.42 mg of liver microsomal protein per ml. In the control reactions with diclofenac (Enamine) and propanolol (Sigma-Aldrich) the NADPH-cofactor system was substituted with phosphate buffer. Dissolved Nec-1f (2 µM, final solvent concentration 1.6 %) was incubated with microsomes at 37 °C, shaking at 100 rpm. Each reaction was performed in duplicates. Five time points over 40 min were analyzed. The reactions were stopped by adding ten volumes of 90% acetonitrile-water (Sigma-Aldrich) to incubation aliquots, followed by protein sedimentation by centrifuging at 14.000×$g$ for 3 min. Supernatants were analyzed using the HPLC system coupled with tandem mass spectrometer (also see Supplementary Table 3 for details). The elimination constant ($k_{\text{el}}$), half-life ($t_{1/2}$), and intrinsic clearance ($\text{Cl}_{\text{int}}$) were determined in the plot of ln(AUC) versus time, using linear regression analysis:

$$k_{\text{el}} = -\text{slope} \tag{1}$$

$$t_{1/2} = \frac{0.693}{k} \tag{2}$$

$$\text{Cl}_{\text{int}} = \left( \frac{0.693}{t_{1/2}} \right) \frac{\mu\text{l}_{\text{incubation}}}{\text{mg}_{\text{microsomes}}} \tag{3}$$

The intrinsic clearance classification bands are then calculated according to the well-stirred model equation

$$\text{Cl}_{\text{int}} = \frac{\text{Cl}_{\text{H}}}{f_u(1 - E)} \tag{4}$$

where $\text{Cl}_{\text{H}}$ is a hepatic clearance (ml/min/kg), $\text{Cl}_{\text{H}} = EQ_{\text{H}}$
$Q_{\text{H}}$ = liver blood flow (mlmin/kg)2
$E$ = extraction ratio, assumed at 0.3 for low clearance and at 0.7 for high clearance compounds
$f_u$ = fraction unbound in plasma, assumed at 1.
The $\text{Cl}_{\text{int}}$ classification values were calculated for mouse, rat, and human species using the literature data on liver weight and microsomal protein concentration[40,41] and are represented in the following table.
The intrinsic clearance groups for the classification of test compounds are shown below.

| Classification group | Intrinsic clearance (µl/min/mg protein) | | |
|---|---|---|---|
| | Mouse | Rat | Human |
| Low clearance | < 8.6 | < 13 | < 8.8 |
| High clearance | > 48 | > 72 | > 48 |

**Assessment of stability in human plasma**. To determine the stability of the Nec-1f in human plasma (plasma stability assay was carried by ENAMINE Ltd., Ukraine), Nec-1f and reference compounds were analyzed and compared at five time points over 120 min using HPLC-MS/MS. Plasma stability is defined as the percentage of parent compound remaining in plasma over the time. Here, incubations were carried out in 5 aliquots of 70 µl each (one for each time point), in duplicates. Nec-1f and respective controls (1 µM, final DMSO concentration 1%) were incubated at 37 °C with shaking at 100 rpm. Five time points over 120 min have been analyzed. The reactions were stopped by adding 420 µl of acetonitrile-water mixture (90:10) with subsequent plasma proteins sedimentation by centrifuging at $14.000 \times g$ for 5 min. Supernatants were analyzed by the HPLC system coupled with tandem mass spectrometer. The percentage of the test compounds remaining after incubation in plasma and their half-lives ($T_{1/2}$) was calculated.

**In vitro off-target pharmacology study and IDO assay**. Potential binding partners of Nec-1f at a concentration of 10 µM were assessed by Eurofins Discovery Services, Celle L'Evescault, France. Compound binding to recombinant proteins was calculated as inhibition (in percent) of the binding of a radioactively labeled ligand specific for each target according to the Eurofins standard protocol. To determine the inhibitory activity of Nec-1f on indoleamine 2,3-dioxygenase (IDO), the ability of IDO to mediate oxidative cleavage of L-tryptophan to N-formyl-kynurenine was measured. Following pre-incubation of E. coli-derived human IDO protein (20 nM) for 15 min at 25 °C with increasing concentrations of 0.1, 1, and 10 µM Nec-1f, 0.1 fM L-tryptophan (L-Trp) in the presence of 20 mM ascorbate, 3.5 µM methylene blue and 0.2 mg/ml catalase in 50 mM potassium phosphate buffer (pH 6.5) was added and incubated for 2 h at 25 °C. The formation of N-formyl-kynurenine was quantified by following the absorbance increase at 321 nm.

**Mice**. All mice used in this study were kept under stable 12-h circles of darkness and light in the respective facilities. Room temperature was kept between 20 and 24 °C and air humidity between 45 and 65% as documented in daily controls. If not otherwise indicated, all cages were IVCs which fulfilled at least Euonorm type II. Mice had access to sterilized standard pellet food and water ad libidum. All cages and nestlets were sterilized by autoclaving before use.

8- to 12-week-old mice were co-housed with 2−5 mice/cage in our facility at the Medizinisch-Theoretisches Zentrum (MTZ) at the Medical Faculty of the Technische Universität of Dresden (TU Dresden). Matching wild-type mice (C57Bl/6N) were initially provided by Charles River, Sulzfeld, Germany, at the age of 6−7 weeks. Gpx4$^{cys/−}$ mice were described previously[42]. In brief, the selenocysteine of GPX4 was genetically changed to a cysteine, resulting in compromised GPX4-activity. To drive the Gpx4 mutation-initiating Cre, mice received an intraperitoneal dose of tamoxifen (2 mg in 100 µl SMOFlipid, HEXAL) at days 1 and 3, whereas the experiments were performed from day 15 on. Mice not harboring the Cre recombinase and mice not containing the Gpx4 mutation also received tamoxifen at days 1 and 3 and were considered controls. Specific information regarding the Fsp1-deficient mice (B6.129-Aifm2$^{tm1Marc}$/Ieg) is available at http://www.informatics.jax.org/strain/MGI:5905052[19]. All experiments were conducted in a strictly blinded manner regarding genotypes and small molecules. All experiments were performed according to German animal protection laws and were approved by German ethics committees and local authorities in Kiel (ethics committee of the CAU Kiel and the Umwelt- und Landwirtschaftsministerium Schleswig-Holstein, respectively) and Dresden (ethics committee of the TU Dresden and the Landesdirektion Sachsen, respectively). Sex-matched, 6−10-week-old C57BL/6 (The Jackson Laboratory, Bar Harbor, ME) and C57BL/6 LysM-GFP mice (provided by M. Miller, Washington University in St. Louis and originally obtained from K. Ley, La Jolla Institute for Allergy and Immunology, La Jolla, CA) were utilized for intravital imaging of transplanted hearts. Heart transplant experiments were approved by the Institutional Animal Studies Committee at Washington University.

**Isolation of primary murine renal tubules**. Primary murine renal tubules were isolated following a modified previously established protocol[14]. In detail (demonstrated in Fig. S7), murine kidneys were removed, washed with PBS, decapsulated, and sliced in four to five slices. Kidney slices of each kidney were transferred in 2 ml reaction tube containing 2 mg/ml collagenase type II in incubation solution (48 µg/ml trypsin inhibitor, 25 µg/ml DNAse I, 140 mM NaCl, 0.4 mM KH$_2$PO$_4$, 1.6 mM K$_2$HPO$_4 \cdot$ 3 H$_2$O, 1 mM MgSO$_4 \cdot$ 7 H$_2$O, 10 mM CH$_3$COONa $\cdot$ 3 H$_2$O, 1 mM α-ketoglytarate and 1.3 mM Ca-gluconate) and digested for 5 min at 37 °C, 850 rpm. Due to the presence of damaged tubules, the first resulting supernatant was

discarded and 1 ml of 2 mg/ml collagenase type II in incubation solution was added to the kidney slices and digested for 5 min at 37 °C, 850 rpm. The supernatant was collected and transferred in a 2 ml reaction tube containing 1 ml ice-cold sorting solution (0.5 mg/ml bovine albumin in incubation solution). The reaction tubes were left on ice for the tubules to precipitate. The supernatant was removed and the tubules were washed twice with ice-cold incubation solution. Once the tubules precipitated the supernatant was removed and the ice-cold sorting solution was added (the volume was adjusted depending on the number of samples needed for the experiment). Tubules were distributed in a twenty-four-well plate containing Dulbecco's Modified Eagle Medium F-12 Nutrient Mixture without phenol red (DMEM/F12, Thermo Fisher), supplemented with 0.01 mg/ml recombinant human insulin, 5.5 µg/ml human transferrin, 0.005 µg/ml sodium selenite (Na$_2$SeO$_3$) (ITS without linoleic acid, Sigma Aldrich), 50 nM hydrocortisone, 100 U/ml penicillin, and 100 µg/ml streptomycin (Pen/Strep, Thermo Fisher).

Generation, cell culture, and induction of cell death in primary murine tubular cells. Murine tubular cells were generated by outgrowth from isolated renal tubules (see above). Primary murine tubules were placed in six-well plates containing Dulbecco's Modified Eagle Medium F-12 Nutrient Mixture without phenol red (DMEM/F12, Thermo Fisher), supplemented with 0.01 mg/ml recombinant human insulin, 5.5 µg/ml human transferrin, 0.005 µg/ml Na$_2$SeO$_3$ (ITS without linoleic acid, Sigma Aldrich), 50 nM hydrocortisone, 100 U/ml penicillin, and 100 µg/ml streptomycin (Pen/Strep, Thermo Fisher). After two days, the outgrown primary murine tubular cells were washed with PBS and a fresh medium was added. When the number of primary murine tubular cells reached approximately 50% confluence, the cells were removed from the six-well plates using Trypsin-EDTA (Thermo Fisher), washed with medium and a Ficoll Paque Plus (Sigma Aldrich) gradient was performed to remove any dead cells, and debris from the murine renal tubules. Cells were washed twice with PBS and seeded in six-well plates with medium. Once the number of grown cells reached approximately 50% confluence, they were washed with PBS and treated with type 2 FIN: RSL3 (also see Supplementary Table 2 for details). After 14 h, medium and cells were collected and prepared for LDH release assay. Images of the treated primary murine tubular cells were obtained using a 20×/0.30 PH1 objective on a Leica DMi1 microscope.

**Induction of cell death in murine renal tubules**. All of the experiments performed contain a negative control to assess LDH release at 0 h of incubation as a quality control. No more than 10% LDH release in these negative controls was tolerated. In these tubules, necrotic cell death occurs spontaneously as demonstrated, but can be accelerated by the addition of ferroptosis-inducing agents such as RSL3. Isolated murine renal tubules were placed in twenty-four-well plates containing the respective agents diluted in Dulbecco's Modified Eagle Medium F-12 Nutrient Mixture without phenol red (DMEM/F12, Thermo Fisher), supplemented with 0.01 mg/ml recombinant human insulin, 5.5 µg/ml human transferrin, 0.005 µg/ml Na$_2$SeO$_3$ (ITS without linoleic acid, Sigma Aldrich), 50 nM hydrocortisone, 100 U/ml penicillin, and 100 µg/ml streptomycin (Pen/Strep, Thermo Fisher). After the indicated time points, the medium of each well was collected and tubules were prepared for an LDH release assay. Images of the treated murine renal tubules were obtained using a 20x/0.30 PH1 objective on a Leica DMi1 microscope.

**Cisplatin-induced acute kidney injury (CP-AKI)**. As reported earlier[43], female mice were rigorously matched for weight, age, and genetic background. Each mouse received a single dose of cisplatin (20 mg/kg) intraperitoneally in a total volume of 400 µl PBS. Following injection, mice were returned into cages (2−5 animals per cage) and monitored closely. After 48 h, retro-orbital blood collection was performed and mice were sacrificed by cervical dislocation. The right kidney was put in 4% normal buffered formalin for 24 h and then transferred to 70% ethanol for storage at room temperature. Correspondingly, the left kidneys were sorted into an Eppendorf tube and shock frozen in liquid nitrogen.

**Bilateral kidney ischemia and reperfusion injury model (IRI)**
Surgical protocol. All male mice were strictly matched for weight, age, and genetic background. Bilateral kidney ischemia and reperfusion injury (IRI) was performed as described in detail earlier[43]. In essence, 15 min prior to surgery, all mice received 0.1 µg/g body weight buprenorphine-HCl intraperitoneally for analgesia. Anesthesia was induced by the application of 3 l/min of volatile isoflurane with pure oxygen in the induction chamber of a COMPAC5 (VetEquip, the Netherlands) small animal anesthesia unit. After achieving a sufficient level of narcosis, typically within 2 min, mice were placed in a supine position on a temperature-controlled self-regulated heating system calibrated to 38 °C and fixed with stripes at all extremities. Anesthesia was reduced to a maintenance dose of 1.5 l/min isoflurane. Breathing characteristics and levels of analgesia were closely assessed visually. The abdomen was opened layer-by-layer to create a 2 cm wide opening of the abdomen. Blunt retractors (Fine Science Tools (FST), Germany) were placed for convenient access. The caecum and gut were carefully mobilized and placed to the left side, where they were placed on a PBS-soaked sterile gauze. The second piece of PBS-soaked gauze was used to sandwich the gut, deliberately lifting the duodenum to visualize the aorta abdominalis. A cotton bud was used to gently push the liver cranially to fully access the right renal pedicle. With the use of a surgical microscope (Carl Zeiss, Jena, Germany), sharp forceps were used to pinch retroperitoneal

holes directly cranially and caudally in the renal pedicle. Via this access, a 100 g pressure micro serrefine (FST 18055-03) was placed on the pedicle to induce ischemia and a timer was started. The cotton bud was removed, and the packed gut switched to the right side of the mouse to visualize the left renal pedicle. If required, the cotton bud was used to gently push aside the spleen or stomach. Once again, retroperitoneal access was achieved by pinching holes with sharp forceps and another 100 g pressure micro serrefine was placed. The time between the placement of both serrefines was recorded (typically 40 s, controlled in all cases to under 1:00 min), the gut was returned into the abdominal cavity and the opening was covered with the two gauze pieces. Twenty-nine minutes after initially starting the timer, the retractors were put in place again and the gut again mobilized and packed to visualize the right kidney. After exactly 30 min (1 s tolerance), the vascular clamp was removed and the gut switched to the right side. After the recorded time difference, this clamp was removed as well. Reperfusion was determined visually for both sides before the gut was returned into the abdominal cavity. The parietal peritoneum and the cutis, respectively, were closed separately by continuous seams using a 6-0 monocryl thread (Ethicon). Isoflurane application was stopped immediately thereafter and 1 mL of pre-warmed PBS was administered intraperitoneally to compensate for any possible dehydration during surgery and to control for potential leakiness of the seams. The mice were divided into pairs of two and put back into the cages. 0.1 µg/g buprenorphine-HCl was administered every 8 h for analgesia. After a 48-h observation period, blood was collected by retroorbital puncture and the mice were sacrificed by neck dislocation. The right kidney was removed to be fixed for 24 h in 4% normal buffered formalin and transferred to 70% ethanol for storage at room temperature. The left kidney was removed and shock frozen in liquid nitrogen before transfer to −80 °C for storage.

*Moderate and severe models of IRI.* In this study, we applied two different doses of ischemia, a severe and a moderate model. The time of ischemia before the onset of reperfusion in the model of moderate IRI was 30 min (Fig. 1B–I, Fig. S2A–D, and Fig. 6A–I) and in the model of severe IRI was 38 min (Figs. 1A and 6J), respectively. Otherwise, the two models are identical.

**Acute oxalate nephropathy.** C57BL/6N mice were procured from Charles River Laboratories (Sulzfeld, Germany) and co-housed in groups of four in filter top cages with unlimited access to food and water. Cages, nestlets, food, and water were sterilized by autoclaving before use. Mice received a single intraperitoneal (i.p.) injection of 100 mg/kg sodium oxalate (Santa Cruz Biotechnology) and 3% sodium oxalate in drinking water and kidneys were harvested after 24 h. As a therapeutic strategy, mice received a single dose of 10 mg/kg Nec-1f or vehicle control (10% DMSO and PBS) 30 min before sodium oxalate injection. Blood and kidneys were collected at sacrifice by cervical dislocation. The middle section of each kidney was fixed for 24 h in formalin to be embedded in paraffin for histological analysis. Intrarenal crystal precipitations were quantified by Pizzolato staining[44]. The experimental procedure was approved by the Regierung von Oberbayern, München, Germany (ROB-55.2-2532.Vet_02-18-127).

**Intravital imaging of transplanted hearts.** Hearts harvested from wild-type C57BL/6 mice were transplanted into the right neck of syngeneic LysM-GFP hosts, as previously described[36]. Recipient mice received no treatment or were treated with Nec-1f (2 mg/g body weight intraperitoneally at the start of the heart transplant procedure). Intravital two-photon imaging was performed using a custom-built microscope running ImageWarp version 2.1 acquisition software (A&B Software). For time-lapse imaging of cell trafficking, we averaged 15 video-rate frames (0.5 s per slice) during the acquisition. Each plane represents an image of 220 × 240 µm in the *x* and *y* dimensions and twenty-one sequential planes were acquired in the *z* dimension (2.5-µm each). Each individual neutrophil was tracked from its first appearance in the imaging window and followed up to the time point at which it dislocated more than 20 µm from its starting position. Fifty microlitres PBS containing 15 µl 655-nm nontargeted Q-dots (Thermo Fisher Scientific) were injected prior to imaging to visualize coronary vessels and assess whether neutrophils were intravascular or extravascular. The number of extravascular neutrophils was divided by the sum of intravascular and extravascular neutrophils to calculate extravasation. Imaris (Bitplane) was used for multidimensional rendering and manual cell tracking.

**Immunohistology and semi-quantitative scoring.** Organs were dissected as indicated in each experiment and put in 4% (vol/vol) neutral-buffered formaldehyde, fixated for 24 h, and then transferred to 70% ethanol for storage. For histologies, the kidneys were dehydrated in a graded ethanol series and xylene, and finally embedded in paraffin. Paraffin sections (3–5 µm) were stained with periodic acid–Schiff (PAS) reagent, according to standard routine protocol. Stained sections were analyzed using an Axio Imager microscope (Zeiss) at 100×, 200×, and 400× magnification. Micrographs were digitalized using an AxioCam MRm Rev. 3 FireWire camera and AxioVision ver. 4.5 software (Zeiss). Organ damage was quantified by two experienced pathologists in a double-blind manner on a scale ranging from 0 (unaffected tissue) to 10 (most severe organ damage).

For the scoring system, tissues were stained with PAS, and the degree of morphological involvement in renal failure was determined using light microscopy. The following parameters were chosen as indicative of morphological damage to

the kidney after ischemia-reperfusion injury (IRI): brush border loss, red blood cell extravasation, tubule dilatation, tubule degeneration, tubule necrosis, and tubular cast formation. These parameters were evaluated on a scale of 0–10, which ranged from not present (0), mild (1–4), moderate (5 or 6), severe (7 or 8), to very severe (9 or 10). Each parameter was determined on at least five different animals.

For 4-HNE stainings (also see Supplementary Table 1 for details), the slides were blocked using 3% $H_2O_2$ followed by antigen retrieval using target retrieval solution (Dako Deutschland GmbH) and boiling in a pressure cooker for 2.5 min. Afterward, sections were blocked with avidin-biotin blocking kit (Vector laboratories) and for 30 min with 5% skim milk (Bio-Rad Laboratories GmbH) and 20% normal goat serum (Vector Laboratories, Burlingame) in 50 mM Tris, pH 7.4. After blocking sections were incubated overnight with polyclonal rabbit anti-4-Hydroxynonenal antibody (Abcam) diluted in 1% BSA in 50 mM Tris, pH 7.4 at 4 °C. Primary antibody was detected using biotinylated goat anti-rabbit secondary antibody, ABC-kit, and ImmPACT-DAB as substrate (all from Vector Laboratories). After counterstaining with hemalaun (Merck KGaA, Darmstadt, Germany) sections were examined using a light microscope (Zeiss Axio Imager.A2, Zeiss GmbH). For negative controls, the primary antibody was omitted and replaced with a blocking solution. At least 12 fields of vision (at 200× magnification) per cross-section from each kidney were evaluated and the degree of 4-HNE-positive tubuli was analyzed using a semi-quantitative score ranging from 0-3: 0 = no staining; 1 = weak granular staining in 1–2 tubules per field of vision; 2 = clear granular and weak diffuse tubular staining in more than 2 tubuli per field of vision; 3 = strong granular and diffuse staining. Mean scores from all analyzed field of visions were calculated.

The tubular injury was scored by detecting the percentage of necrotic structures, dilated tubules, casts and cellular infiltrates per HPF. To do so a grid with a raster size of 2.5% of the total image area was imposed on top of each image and the number of squares covering either of the histopathological features was counted individually or as a composite (summary score). Pizzolato's staining visualized calcium oxalate crystals and crystal deposit formation in the kidney was evaluated the following: no deposits = 0 point; crystals in papillary tip = 1 point; crystals in cortical medullary junction = 2 points; crystals in cortex = 3 points. When crystals were observed in multiple areas, the points were combined.

A TUNEL kit (Cell death detection kit, Roche) was used to quantify the number of cells with DNA strand breaks per HPF. The green fluorescent area was measured and the number of stained cells per slide was counted.

**Pharmacokinetic analysis of Nec-1f in mice (tissue PK studies).** This study was initiated to determine the pharmacokinetic characteristics of the Nec-1f in mice following i.p. administration. Experiments were conducted by Enamine, Ltd, Ukraine. Levels of the Nec-1f were determined by LC-MS/MS in the blood plasma, whole blood, brain, liver, kidney, heart, lung, and pancreas over time after a single dose. 8−9 weeks old male C57BL/6 mice were used in this study. Mice were randomly assigned to treatment groups before the pharmacokinetic study. Mice were fasted for 3–4 h before dosing. Nine time points (5, 15, 30, 60, 120, 240, 360, 480, and 1.440 min) were set for the pharmacokinetic study. Groups of three mice were used for each time point. Nec-1f was administered at 2 mg/ml.

Mice were i.v. injected with 2,2,2-tribromoethanol (Sigma-Aldrich) at the dose of 150 mg/kg prior to blood collection from the orbital sinus in microtainers containing $K_2EDTA$ (150 µl of whole blood was taken into a 1.5 ml tube, the residue was centrifuged at $5.500 \times g$) and sacrificed by cervical dislocation. Tissues samples (brain (left lobe), left kidney, liver (large lobe fragment), lungs (both), pancreas, heart) were processed immediately, snap-frozen, and stored at −70 °C for subsequent analysis.

Plasma (40 µl) and tissue samples (weight 100 ± 1 mg) were homogenized with 400 µl of the internal standard solution IS400 using steel or zirconium oxide beads (115 ± 5 mg) in The Bullet Blender® homogenizer for 30–120 s and centrifuged for 4 min at $14.000 \times g$. Two microliters of each sample were injected into LC-MS/MS system. Diclofenac (400 ng/ml in water−methanol mixture 1:9, v/v) was used as a standard for quantification of Nec-1f in plasma, blood, liver, heart, lung, pancreas, and kidney as well as in brain (400 ng/ml in water-methanol mixture 1:9, v/v) samples. Concentrations of Nec-1f were then determined using HPLC-MS/MS using the following conditions: column: Phenomenex Luna C18(2) (50 × 2 mm, 5 µm), mobile phase A: acetonitrile (Sigma-Aldrich): water: formic acid (Sigma-Aldrich) = 50: 950: 1, mobile phase B: acetonitrile: formic acid = 100: 0.1, linear gradient: 0 min 0% B, 0.9 min 100% B, 1.1 min 100% B, 1.11 min 5% B, 2.8 min stop, elution rate: 400 µL/min. A divert valve directed the flow to the detector from 1.65 to 2.25 min, column temperature: 30 °C, scan type: positive MRM, ion source: turbo spray, ionization mode: ESI, nebulize gas: 15 l/min, curtain gas: 8 L/min, collision gas: 4 l/min, ionspray voltage: −4.200 V, temperature: 400 °C.

Calibration standards for analysis of Nec-1f in plasma, blood, and tissue samples were established as follows: The stock solution was consecutively diluted with IS400 to get a series of calibration solutions with final concentrations of 10.000, 5.000, 2.500, 1.000, 500, 200, 100, 50, 20, 10, 5, and 2 ng/ml. Where necessary, samples were homogenized with 400 µl of corresponding calibration solution using zirconium oxide beads (115 ± 5 mg) in The Bullet Blender® homogenizer for 30 s at speed 8. Calibration curves were constructed using blank samples of the respective tissue.

To obtain calibration standards, blank samples were mixed with 200−400 µl of corresponding calibration solution. After mixing by pipetting and centrifugation for 4 min at 15.000 × g, 2 µl of each supernatant was injected into LC-MS/MS system.

The pharmacokinetic data analysis was performed using noncompartmental, bolus injection, or extravascular input analysis models in WinNonlin 5.2 (PharSight). Data below the lower limit of quantitation (LLOQ) were presented as missing to improve the validity of $t_{1/2}$ calculations. Measurements were performed using Shimadzu Prominence HPLC (also see Supplementary Table 3 for details) system including vacuum degasser, gradient pumps, reverse phase HPLC column, column oven, and autosampler. Mass spectrometric analysis was performed using an API 4.000 QTRAP mass spectrometer from Applied Biosystems/ MDS Sciex (AB Sciex) with Turbo V ion source and TurboIonspray interface. The TurboIonSpray ion source was used in both positive and negative ion modes. The data acquisition and system control were performed using Analyst 1.6.3 software from AB Sciex (Canada).

**Statistical analysis**. Statistical analyses were performed using Prism 8 (GraphPad software, San Diego, CA, USA). To test the null hypothesis of no difference between groups in the survival experiments we plotted the animals in a Kaplan-Meier curve and used the log-rank test for statistics. Mouse data on damage scores and serum levels of creatinine and urea were analyzed by a two-tailed parametric t-test. In all other experiments, a two-sided t-test was used for normally distributed data sets, and one-way ANOVA followed by post hoc Tukey's multiple comparisons test was used for analyzing data from FACS analysis to compare multiple treatments. Data were considered significant when * $p < 0.05$, ** $p < 0.01$, *** $p < 0.001$, **** $p < 0.0001$.

**Reporting summary**. Further information on research design is available in the Nature Research Reporting Summary linked to this article.

## Data availability

Source data has been made available for this paper. Any additional data generated supporting the findings of this study will be made available from the corresponding author upon reasonable request. Source data are provided with this paper.

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

## Acknowledgements

We would like to cordially thank Cemile Jakupoglu and Markus Brielmeier for help in generating the *Fsp1*$^{-/-}$-mice, Romy Opitz for expert technical assistance, and Renata Drtina for critically reading the manuscript. We thank the histology facility, the FACS

facility, as well as the light microscopy facility at the CMCB Dresden, in particular Hella Hartmann, for the expert help. Work in the Linkermann Lab is funded by the Medical Clinic 3, University Hospital Carl Gustav Carus Dresden, Germany, and supported by the SFB-TRR 205, SFB-TRR 127, SPP2306, and the international research training group (IRTG) 2251. This work was supported by the German Research Foundation (DFG) (Heisenberg-Professorship to A.L. (project number 324141047), GE2845/1-1 to F.G., AN372/24-1 to H.J.A. and KR3363/3-1 to N.K.). We received further funding for this project from the transCampus initiative of S.R.B. Work in the Conrad Lab is funded by the DFG (CO 291/5-2; CO 291/7-1; PR 1752/3-1), the German Federal Ministry of Education and Research (BMBF) VIP + program 03VP04260, the Helmholtz Validation Fund (0042), the Ministry of Science and Higher Education of the Russian Federation (075-15-2019-1933), the Else Kröner-Fresenius-Stiftung, the Bavarian Ministry of Economic Affairs, Regional Development and Energy (StMWi), and M.C. has further received funding from the European Research Council (ERC) under the European Union's Horizon 2020 research and innovation program (grant agreement No. GA 884754). K.A. and C.D. received funding by the SFB 1350 TP C2.

## Author contributions

W. T., C. M., C. S., A. B., A. v. M., N. Z. G., F. G. N. H., F. M., M. L. S. L., C. D., J. U. B., S. D., I. I., J. M., W. L., S. S., and M. G. performed experiments. B. P., N. K., R. K., S. M., P. J. H., H.-J. A., D. K., and M. C. provided essential reagents and/or genetically modified mice that made this study possible and helped to design experiments. K. A., S. R. B., C. H., M. G., D. P., and A. L. designed the experiments. A. L. wrote the paper.

## Funding

## Competing interests

A. L., C. S, and M. G issued a patent for Nec-1f with the number 20160943.5. M. C. is co-founder and shareholder of ROSCUE Therapeutics GmbH. Apart from this, all authors declare no conflict of interest regarding any of the presented data in this paper.

## Additional information

[1]Division of Nephrology, Department of Internal Medicine 3, University Hospital Carl Gustav Carus at the Technische Universität Dresden, Dresden, Germany. [2]Biotechnology Center, Technische Universität Dresden, Dresden, Germany. [3]Pharmaceutical Institute, Pharmaceutical Chemistry I, University of Bonn, Bonn, Germany. [4]Institute of Physiology, Christian-Albrecht-University Kiel, Kiel, Germany. [5]Division of Nephrology, Department of Medicine IV, University Hospital LMU Munich, Munich, Germany. [6]Department of Surgery, Washington University, Saint Louis, MO, USA. [7]Department of Chemistry and Biomolecular Sciences, University of Ottawa, Ottawa, ON, Canada. [8]Institute of Metabolism and Cell Death, Helmholtz Zentrum München, Neuherberg, Germany. [9]Department of Nephropathology, Friedrich-Alexander University (FAU) Erlangen-Nürnberg, Erlangen, Germany. [10]Clinic for Renal and Hypertensive Disorders, Rheumatological and Immunological Disease, University Hospital of the RWTH Aachen, Aachen, Germany. [11]Department of Internal Medicine, Nephrology and Transplantation, Erasmus Medical Center, Rotterdam, The Netherlands. [12]Department of Pathobiology, University of Illinois, Urbana, IL, USA. [13]Department of Internal Medicine 3, University Hospital Carl Gustav Carus at the Technische Universität Dresden, Dresden, Germany. [14]Diabetes and Nutritional Sciences, King's College London, London, UK. [15]Center for Regenerative Therapies, Technische Universität Dresden, Dresden, Germany. [16]Paul Langerhans Institute Dresden of Helmholtz Centre Munich at University Clinic Carl Gustav Carus of TU Dresden Faculty of Medicine, Dresden, Germany. [17]Lee Kong Chian School of Medicine, Nanyang Technological University, Singapore City, Singapore. [18]Institute of Pathology, University Hospital of Cologne, Cologne, Germany. [19]Department of Pathology and Immunology, Washington University, Saint Louis, MO, USA. [20]National Research Medical University, Laboratory of Experimental Oncology, Moscow, Russia. [21]These authors contributed equally: Wulf Tonnus, Claudia Meyer. ✉email: andreas.linkermann@ukdd.de

