## [Peer Review File · Nature Communications]

REVIEWER COMMENTS

Reviewer #1 (Remarks to the Author):

As this reviewer pointed out in the initial review comment, this is an important and timely study in ferroptosis field. During the revision, the authors have performed extensive work to address review comments, including providing drug concentration information and statistical analyses. I also agree with the authors' discussion on their animal studies. Therefore, I endorse the publication of this manuscript.

Reviewer #2 (Remarks to the Author):

One issue that I believe needs to be addressed before the manuscript can be published is the reference to Nec-1f as a dual inhibitor of necroptosis and ferroptosis. Nec-1f works similarly to Nec-1s by inhibiting RIPK1 kinase activity, therefore it should not be described as an inhibitor of necroptosis but rather as an inhibitor of RIPK1. This is important as RIPK1 inhibition is not equivalent with necroptosis inhibition. First, RIPK1 induces both apoptosis and necroptosis, which will be both inhibited by Nec-1f. Second, there are RIPK1-independent pathways driving necroptosis that will not be affected by Nec-1f. Therefore referring to Nec-1f as a dual inhibitor of necroptosis and ferroptosis is misleading.

Reviewer #3 (Remarks to the Author):

In the revised version of the manuscript, now submitted to Nature Communications, the authors have addressed several issues that were raised by all three reviewers. I believe it is very important that they properly qualified this reagent as a tool compound of unknown target for ferroptosis and of unknown biochemical potency regarding RIPK1 inhibition. While they acknowledge in their response to reviewers that Nec-1f is a solid inhibitor of necroptosis and weak inhibitor of ferroptosis, this is not reflected in the Abstract or in the last sentence of the Introduction. The authors should adjust the text in these two sections to reflect these important facts.

In response to Reviewer 2 regarding the reluctance of the authors to check Nec-1f in the TNF mediated SIRS model, I disagree with their answer. There are many reports of testing RIPK1 inhibitors in this model (and several of them using inhibitors with much higher selectivity and potency against RIPK1): Harris et al, ACS Med Chem 2013; Berger et al, CDDiscov 2015; Harris et al, J Med Chem 2017; Patel et al, CDD 2020; Riebeling et al, CDD 2021. In addition, MLKL KO does provide a partial protection in the TNF SIRS model (Massenhausen, CDDis 2018), while Casp-8/RIPK3 or Casp-8/MLKL DKO mice are fully protected (Greve et al, Front Cell Inf Micro 2017; Newton et al, CDD 2016) suggesting that inhibition of RIPK1 by specific inhibitors blocks necroptosis and apoptosis as Reviewer 2 stated. The authors should modify their data interpretation and conclusions accordingly. The western blot data in figure 2G still needs explanation as it is clear that Fer-1 strongly potentiates

RIPK1 phosphorylation compared to DMSO treatment. Why is that? Does Fer-1 inhibit caspases or does it have some unknown way of boosting RIPK1 phosphorylation?

Response to Referees

We would like to start this response by pointing out how much we appreciate the comments of all referees and the editor. We respond to the comments by the referees in **bold writing**. We addressed every comment in a point-by-point manner.

Reviewer #1

As this reviewer pointed out in the initial review comment, this is an important and timely study in ferroptosis field. During the revision, the authors have performed extensive work to address review comments, including providing drug concentration information and statistical analyses. I also agree with the authors' discussion on their animal studies. Therefore, I endorse the publication of this manuscript.

We thank this reviewer for the positive feedback on our work.

Reviewer #2

One issue that I believe needs to be addressed before the manuscript can be published is the reference to Nec-1f as a dual inhibitor of necroptosis and ferroptosis. Nec-1f works similarly to Nec-1s by inhibiting RIPK1 kinase activity, therefore it should not be described as an inhibitor of necroptosis but rather as an inhibitor of RIPK1. This is important as RIPK1 inhibition is not equivalent with necroptosis inhibition. First, RIPK1 induces both apoptosis and necroptosis, which will be both inhibited by Nec-1f. Second, there are RIPK1-independent pathways driving necroptosis that will not be affected by Nec-1f. Therefore, referring to Nec-1f as a dual inhibitor of necroptosis and ferroptosis is misleading.

We entirely agree with the referee and have now referred to Nec-1f as an inhibitor of RIP1 kinase activity and ferroptosis. We agree that RIPK1 inhibition is not equivalent with necroptosis. We hope we could clarify the confusion.

Reviewer #3

In the revised version of the manuscript, now submitted to Nature Communications, the authors have addressed several issues that were raised by all three reviewers. I believe it is very important that they properly qualified this reagent as a tool compound of unknown target for ferroptosis and of unknown biochemical potency regarding RIPK1 inhibition.

We thank the referee for the positive feedback on our revision.

While they acknowledge in their response to reviewers that Nec-1f is a solid inhibitor of necroptosis and weak inhibitor of ferroptosis, this is not reflected in the Abstract or in the last sentence of the Introduction. The authors should adjust the text in these two sections to reflect these important facts.

Done.

In response to Reviewer 2 regarding the reluctance of the authors to check Nec-1f in the TNF mediated SIRS model, I disagree with their answer. There are many reports of testing RIPK1 inhibitors in this model (and several of them using inhibitors with much higher selectivity and potency against RIPK1): Harris et al, ACS Med Chem

2013; Berger et al, CDDiscov 2015; Harris et al, J Med Chem 2017; Patel et al, CDD 2020; Riebeling et al, CDD 2021. In addition, MLKL KO does provide a partial protection in the TNF SIRS model (Massenhausen, CDDis 2018), while Casp-8/RIPK3 or Casp-8/MLKL DKO mice are fully protected (Greve et al, Front Cell Inf Micro 2017; Newton et al, CDD 2016) suggesting that inhibition of RIPK1 by specific inhibitors blocks necroptosis and apoptosis as Reviewer 2 stated. The authors should modify their data interpretation and conclusions accordingly.

We have now modified our conclusions according to Nec-1f as a RIPK1 inhibitor. We have adapted the abstract, the last sentence of the introduction and the results section. We also thank the referee for referring our own data (von Mäßenhausen et al., CDDis 2018, Greve et al., 2017) in this respect. However, we continue to respectfully disagree on that the protection of Casp8/RIPK3^{dko} or Casp8/MLKL^{dko} mice implies a role for apoptosis. This hypothesis fails to explain why treatment with zVAD-fmk does not protect but rather sensitizes mice (Cauwels, Nat. Immu 2003), also in combination with Nec-1s (Linkermann et al., Molecular Medicine). Simple inhibition of apoptosis and RIPK1 also does not currently explain the partial protection of GSDMD-ko mice in this model (Demarco et al., Science advances 2021).

The western blot data in figure 2G still needs explanation as it is clear that Fer-1 strongly potentiates RIPK1 phosphorylation compared to DMSO treatment. Why is that? Does Fer-1 inhibit caspases or does it have some unknown way of boosting RIPK1 phosphorylation?

We agree with the referee. We added a new section to the results part to reflect on this observation. The section now reads: “Nec-1s and Nec-1f, due to the nature of inhibiting RIPK1 kinase activity, also prevented phosphorylation at S166 in RIPK1 (Fig. 2G). In this experiment, we added Fer-1 as a control compound and found that RIPK1 S166 phosphorylation was increased in the presence of zVAD-fmk, indicating a caspase-independent unknown way of boosting RIPK1 phosphorylation.”